# Lack of Trust, Conspiracy Beliefs, and Social Media Use Predict COVID-19 Vaccine Hesitancy

**DOI:** 10.3390/vaccines9060593

**Published:** 2021-06-03

**Authors:** Will Jennings, Gerry Stoker, Hannah Bunting, Viktor Orri Valgarðsson, Jennifer Gaskell, Daniel Devine, Lawrence McKay, Melinda C. Mills

**Affiliations:** 1School of Economic, Social and Political Sciences, University of Southampton, Southampton SO17 1BJ, UK; w.j.jennings@soton.ac.uk (W.J.); g.stoker@soton.ac.uk (G.S.); h.j.b.willis@soton.ac.uk (H.B.); v.o.valgardsson@soton.ac.uk (V.O.V.); j.gaskell@soton.ac.uk (J.G.); l.a.mckay@soton.ac.uk (L.M.); 2St. Hilda’s College, University of Oxford, Oxford OX4 1DY, UK; daniel.devine@st-hildas.ox.ac.uk; 3Leverhulme Centre for Demographic Science & Nuffield College, University of Oxford, Oxford OX1 1NF, UK

**Keywords:** COVID-19, vaccination, trust, misinformation

## Abstract

As COVID-19 vaccines are rolled out across the world, there are growing concerns about the roles that trust, belief in conspiracy theories, and spread of misinformation through social media play in impacting vaccine hesitancy. We use a nationally representative survey of 1476 adults in the UK between 12 and 18 December 2020, along with 5 focus groups conducted during the same period. Trust is a core predictor, with distrust in vaccines in general and mistrust in government raising vaccine hesitancy. Trust in health institutions and experts and perceived personal threat are vital, with focus groups revealing that COVID-19 vaccine hesitancy is driven by a misunderstanding of herd immunity as providing protection, fear of rapid vaccine development and side effects, and beliefs that the virus is man-made and used for population control. In particular, those who obtain information from relatively unregulated social media sources—such as YouTube—that have recommendations tailored by watch history, and who hold general conspiratorial beliefs, are less willing to be vaccinated. Since an increasing number of individuals use social media for gathering health information, interventions require action from governments, health officials, and social media companies. More attention needs to be devoted to helping people understand their own risks, unpacking complex concepts, and filling knowledge voids.

## 1. Introduction

Governments are rapidly mobilising vaccines against COVID-19 [1], with success relying on sufficient uptake; yet there is a rise in vaccine hesitancy, linked to loss of trust, complacency, and misinformation [2,3]. Trust is crucial to ensuring compliance with public health measures [4,5,6], but governments and experts have needed to communicate uncertain advice, and even reversals in advice, eroding public trust [7]. COVID-19 is not only a pandemic, but an ”infodemic” of complex and dynamic information—both factual and incorrect. This can generate vaccine hesitancy [8], which the WHO listed as one of the top 10 threats to global health in 2019. But who does the public trust, and does trust depend on where the public acquire their information? The growth in Internet use and reliance on social media sources such as YouTube, Facebook, Twitter, and TikTok has changed the landscape of information gathering, with 72% of Americans and 83% of Europeans using the Internet as a source for health information [9]. Conspiracy and anti-vax beliefs and low trust in institutions are associated with a greater reliance on social media for health information, but research on this topic until now has primarily used small, selective samples (e.g., MTurk) [10,11]. In order to empirically inform these urgent issues, we present the results of a survey fielded during the first vaccine rollout in the UK between 12 and 18 December 2020, on a nationally representative sample of 1476 adults, complemented with 5 focus groups conducted during roughly the same period (see Appendix A).

## 2. Background and Hypotheses

Based on previous literature, we test three hypotheses. Trust is confidence in the action of others, mistrust measures vigilance in whether actors or information are trustworthy, and distrust denotes a negative orientation towards institutions or actors [4,5]. A recent survey in England found that those endorsing conspiracy theories were less likely to adhere to government guidelines, and had a general distrust in institutions [12]. Another found a link between trust, conspiratorial beliefs, and vaccine hesitancy [13]. Individuals may not trust the government, but be more willing to ”follow the science” and trust scientific or health experts. In our first hypothesis, we therefore contend that multiple facets of trust are crucial in understanding vaccine uptake [4,7,14]. We hypothesise that trust in government and a positive view of the government’s handling of the crisis will predict higher vaccine willingness, while vaccine distrust/mistrust, and mistrust/distrust of government, predict greater hesitancy (H1). A study in Italy during the initial COVID-19 outbreak found that trust in scientists and health authority experts initially increased, and predicted better knowledge about COVID-19 [15]. More generally, there is evidence that societal-level trust in science is related to vaccine confidence [16]. As a sub-hypothesis, we therefore predict that those with higher levels of trust in health institutions and experts will exhibit higher vaccine willingness [11] (H1.1).

Social trust enables the collective action needed to achieve sufficient population vaccination levels, with previous research demonstrating that social capital is positively associated with health [17]. Since deaths from COVID-19 are concentrated in higher ages and higher risk groups [18], public discourse has been centred around ”vulnerable” groups and herd immunity [19]. Research has shown that this can result in people holding lower perceived personal risk, interpreting risk as only targeting the ”vulnerable” and not related to them personally [3]. If personal risk is perceived as low, this translates into lower vaccination intentions [20,21]. In our second hypothesis, we therefore expect those with higher collective social trust and a higher perceived personal threat from COVID-19 to be less vaccine-hesitant (H2).

A wide body of literature examining a variety of vaccines has shown that holding general conspiracy or COVID-19 misinformation beliefs lowers vaccine willingness [3]. Trust itself is a predictor of susceptibility to misinformation about COVID-19 [22]. The main sources of vaccine misinformation are on social media. An analysis of 1300 Facebook pages during the 2019 measles outbreak found that anti-vax pages grew by 500%, compared to 50% growth of pro-vaccine pages [23]. With social media, individuals can now also more easily find themselves in echo chambers. Once a YouTube user develops a watch history, for instance, a filter bubble tailors their “Top 5” and “Up-Next” recommendations, with watching videos promoting vaccine misinformation leading to more misinformed recommendations [24]. Based on this research, in our third hypothesis, we expect that consumers of social media are more likely to be vaccine-hesitant than consumers of traditional media sources (e.g., TV, newspaper, radio, etc.) (H3). This is especially likely for platforms where algorithms channel future content based on past history, and where content remains relatively unregulated. 

Socio-demographic and political factors are also central to understanding vaccine hesitancy. Based on existing research, we anticipate socio-political demographic variation by digital disparities in information seeking, with younger, more educated, and higher socio-economic status individuals being more active [9,25]. Research has also shown that political conservatives are more likely to believe in vaccine conspiracies [11]. An analysis of popular anti-vax Facebook pages found that the majority (72%) were mothers, often linked to childhood vaccinations for measles, mumps, and rubella (MMR) [26].

## 3. Data and Analytical Methods

### 3.1. Data

We commissioned Ipsos MORI to conduct an online survey of 1476 adults in the UK, 12–18 December 2020, using a quota-controlled selection of preregistered panel members, with population targets set to ensure representativeness of the national population. The fieldwork was conducted shortly after the launch of the UK vaccination programme (8 December), and the survey was designed to investigate factors that impact vaccine uptake or hesitancy. Our measures were also connected to trust across a range of arenas: from trust in government in general (including measures of mistrust and distrust), to trust in experts and information from the media, in addition to distrust in vaccines and general conspiracy beliefs. They also included perceptions of the threat posed by COVID-19 (to people personally, to their jobs/businesses, and to the country), and of how well government was considered to be handling particular aspects of the crisis. We also examined how respondents consumed or shared information, and their use of ”vertical” (e.g., TV, radio, newspaper, etc.) or ”horizontal” (e.g., online, talking to people, etc.) sources for following news about politics or current affairs, as well as their use of specific social media platforms. We likewise collected information on key demographic variables (age, gender, education, social grade, urban/rural, number of children in household), current voting intentions, and whether people had tested positive for, or believed they had been infected with, COVID-19. The full questionnaire with question wording and response options is presented in the SI.

We also ran five focus groups exploring themes of trust and COVID-19 from 30 November to 7 December 2020, with 29 participants across 5 locations in Bristol [2] and Oldham [3] in the UK. These locations were selected as exemplars of a relatively affluent, diverse city in the South of England (Bristol), and a former industrial town in the North of England (Oldham), with groups recruited to reflect particular profiles in terms of age group, social class, and political (Brexit) identity. This ensured that a range of demographics and opinions were represented. A detailed description of the focus group sample is provided in Appendix A. One of the topics we asked about was whether people were willing to be vaccinated. We also probed to what extent they trusted the current government to manage the coronavirus crisis, how much they trusted information from the government, and their views on conspiracy theories and stories circulating about COVID-19, the effectiveness of local lockdowns and the tier system, the balance between minimizing infections and keeping the economy going, and whether a vaccine is the only way the country can get ”back to normal”.

### 3.2. Analytical Methods

For the survey analysis, we measured vaccine willingness by asking “If a vaccine for COVID-19 were available to me, I would get it”, dichotomised into those who strongly agreed or tended to agree (71%) versus those who strongly disagreed or tended to disagree, neither, or were unsure (29%). Of the respondents, 49% strongly agreed they would get the vaccine, 22% indicated they tended to agree they would get it, 11% neither agreed nor disagreed, 7% tended to disagree, and 7% strongly disagreed (with 5% indicating “don’t know”)—as shown in Appendix A. All independent variables were rescaled to a range from 0 to 1 in our statistical analysis, in order to allow for direct comparison of effect sizes—see Appendix A.

We first estimated bivariate logistic regressions of willingness to get the vaccine on our predictors. This enabled us to understand the relationships between each of the measures and vaccine uptake. We then estimated logistic regression models of willingness to get vaccinated, controlling for demographics and partisanship, including blocks of variables in separate multivariate models as a further check and to directly test our main hypotheses. Finally, we estimated a combined logistic regression model, which includes all of our predictors in a single model, offering a stricter test of our hypotheses. We also undertook various sensitivity checks of the models, described below.

For the focus groups, we analysed the pseudonymised transcripts of the groups and coded answers using NVivo software in two waves: A first wave identified participants’ willingness, or lack thereof, to be vaccinated, which we grouped into three categories: willing, unwilling, and hesitant. We then inductively collected every justification offered by the participants in this process and identified recurring themes. The answers were then coded again using these new categories. Through this inductive analytical framework, we analysed the data along three main axes: we looked for areas of consensus within and across the groups around the themes that emerged in participants’ answers; and we juxtaposed participants’ vaccine positions with their evaluations of the government’s handling of the pandemic and their perceptions of the scientific basis for political action throughout the crisis.

More detail on the reported vaccine intentions of the focus group participants by other key factors—including location, social class, whether or not they were furloughed, and their trust profiles—are presented in Appendix A. Of the 29 participants in our focus groups, 14 stated that they would take the vaccine, but 1 not straight away; 11 said that they would not take the vaccine; 4 were unsure.

## 4. Results

### 4.1. Positive Factors for Vaccinaton Willingness

Figure 1 plots the odds of willingness to get the COVID-19 vaccine by variable (with a log scale used for the odds ratio on the x-axis). To interpret these effects, where the odds ratio exceeds 1.0 (marked by the red vertical line), this indicates that the predictor is associated with a greater willingness to be vaccinated, while where it is lower than 1.0, this indicates that it is associated with a lower willingness to receive the vaccine.

Of those factors that were associated with increased likelihood of vaccine willingness, age and trust in health organisations (i.e., the National Health Service and the WHO) had the strongest bivariate associations. The odds ratio of just over 20 means that the oldest respondents were over 20 times more likely to express willingness to get the vaccine compared to the youngest. Note that the age of respondents in our sample ranged from 18 to 87 but, as with all of the variables, this was rescaled to a range from 0 to 1 in our analysis. Similarly, someone with a high level of trust in health organisations was around 20 times more likely to be willing to be vaccinated than someone with the lowest level of trust. The next largest positive association was for people who consume a large amount of information from traditional media, followed by positive evaluations of government handling of the COVID-19 crisis, trust in experts and government, social trust, perceived personal threat from COVID-19, support for the governing Conservative Party, trust in information from the media, those with a graduate degree or above and, finally, those who consume a large amount of information online. 

These factors were also apparent in the focus groups. Those who said they would take the vaccine were more likely to have stated that they trusted the government’s handling of the pandemic. Interestingly, there was acknowledgement in this group of the inconsistencies in this handling, and even references to incompetence, but an implicit (and sometimes explicit) trust that the government are trying their best or to do the right thing. Indeed, those in this group were the ones most likely to mention positive attitudes about the government’s furlough scheme, possibly associating this with benevolence. Similarly, in assessing the balance between protecting lives and supporting the economy, they recognised the difficulty that the government faced.

These vaccine-willing participants were also more likely to see the government as having ”followed the science”, though they were split on whether the virus was a natural occurrence or man-made, with some expressing doubt over the validity of COVID-19 death rates. They seemed to implicitly trust the science and vaccine approval processes, recognising the extraordinary effort that has gone into getting to that point. They also understood the (mRNA—although nobody explicitly mentioned that term) vaccine to be a relatively new kind of technology. There was also broad recognition of the need for the vast majority of people to get vaccinated. The main reasons stated by participants for their decision to have the vaccine were to protect their families and/or as their civic duty to protect society. It was seen as the only way back to some form of normality.

### 4.2. Negative Factors for Vaccinaton Willingness

Of those factors that decrease the likelihood of willingness to get the vaccine in Figure 1, conspiracy beliefs have the largest effect, followed by distrust of vaccines, belief in COVID-19 misinformation, and ”lockdown scepticism” [27]. As described in detail in the SI, our battery of four questions that were designed to measure distrust included: ”vaccine safety data is often made up”, ”people are being lied to about the effectiveness of vaccines”, ”data about the effectiveness of vaccines is often made up”, ”vaccines are not harmful”. General mistrust and distrust in government (based on factor variables formed from our battery of trust, mistrust, and distrust statements [5], detailed in the SI) are associated with odds of being willing to get the vaccine that are around three times lower. Users of Instagram, YouTube, Snapchat, and TikTok are all less likely to express willingness to be vaccinated, as are women.

Information was also important in the focus groups. Those who stated that they would not get vaccinated were more likely to have said COVID-19 when asked what issue the government was least trustworthy on. As justification, they cited the perception that there is ”one rule for us, another for them”, scepticism around reported COVID-19 death figures, and the perceived unfairness (politicisation) and inconsistencies of the tier system of local lockdowns. Participants who declared that they would not get the vaccine pointed to the policy confusion, scandals over PPE (personal protective equipment), schools, the Prime Minister not attending ”COBRA” meetings (crisis response meetings held in the Cabinet Office Briefing Rooms in London), perceived corruption, and policy leaking to newspapers as evidence of generally untrustworthy behaviour. They were also less likely to believe that the government had ”followed the science” (a term that was repeatedly brought up in discussions) throughout. 

A common thread amongst these vaccine unwilling participants was their view that the government put too much emphasis on lockdown measures at the expense of the economy. The participants mentioned the longer term impact of economic fallout on livelihoods, the politicised nature of the tier system—which those in the groups in Oldham saw as punishment for Andy Burnham (the Mayor of Greater Manchester) standing up to the government—and strong favouritism for London over the North. The majority of people who would refuse the vaccine (8 out of 11) either believed that the virus was man-made or were willing to keep an open mind to this possibility. This was because they identified the uneven effects of the virus on different population groups as some sort of targeting that they perceived as unnatural, and as a form of population control. None of them believed that the vaccine was the only way back to normality. In fact, they offered either some adapted understanding of herd immunity, or arguments that the virus was not as deadly as described (linked to scepticism of registered deaths), concluding that most people do not need a vaccine. Similarly, in justifying their decision not to get a vaccine, they highlighted their belief that the vaccine process had been rushed, that not enough testing had been undertaken, and the potential of unknown side effects. One compared it to the thalidomide scandal of the late 1950s as an example of what can go wrong with untested medicine. These assessments did not take into account the fact that in order to be effective in a population, a vaccine needs to be administered to a sufficient percentage of the population. Instead, people believed that those who were most vulnerable to COVID-19 should potentially receive the vaccine, but as they did not personally find themselves in an at-risk category, they believed that they would not need a vaccine. Overall, this group believed that the unknown possible side effects from the vaccines were a greater risk than the possible death or long-term effects of COVID-19.

Those who were unsure about whether they would accept a vaccine were mainly nervous about the rapidity of the vaccine development process, identifying the need for more testing. They did not feel that a vaccine was the only way back to normality, largely attributed to mixed interpretations around the notion of “herd immunity”. This group also expressed a lack of trust in information provided by the government about the crisis, citing inconsistencies in how COVID-19 deaths were recorded as justification. They also expressed scepticism or real uncertainty around theories over the origin of the virus, saying that it was very difficult to know what to believe. Finally, one participant mentioned being hesitant around the vaccine because of the idea that some form of vaccine passport would be required in order to return to normality. Another outlined the different efficacies of the various vaccines, suggesting that the government had purchased more of a less effective vaccine.

### 4.3. Hypothesis Testing

We next consider how these patterns hold, controlling for the propensity of particular demographic groups to be more or less willing to get vaccinated. Figure 2 presents the coefficients from 12 separate logistic regression models including different blocks of independent variables in turn. These broadly confirm the findings from the bivariate regression models, though a few variables lose their statistical significance in these models compared to the bivariate regressions. It is noticeable that attitudinal predictors typically have larger effects than demographic predictors (with the exception of age). The beliefs that individuals hold tend to be a stronger guide as to whether or not they are vaccine-hesitant than their demographic characteristics—an important finding for interventions and policymakers.

Figure 3 and Table 1 present results from a combined logistic regression model that includes all of our predictors in a single model, providing the strictest test of potential effects on vaccine willingness. The results in the table start with a baseline model including our measures of trust, attitudes towards COVID-19, and distrust of vaccines, plus conspiracy beliefs and demographic controls (model 1). Information sources are then added (model 2), followed by social media use (model 3) and online consumption of information (model 4). We focus on the final model for the purpose of analysis. In this comprehensive model, many of the coefficients lose their statistical significance, but the majority of the central findings remain. We find evidence for H1 and H2, along with suggestive findings for H3. For H1, those expressing the highest levels of vaccine distrust are around one-tenth as likely to be willing to get the vaccine as those who have the lowest levels of vaccine distrust, holding all other variables constant. The effect size is not surprising given the proximity to our dependent variable. Those who mistrust government are more hesitant, where (holding the other variables constant) going from the lowest level of mistrust to the highest is associated with being about one-third as likely to be willing to be vaccinated. Those with the highest levels of trust in health institutions are just over six times more likely to express vaccine willingness compared to the lowest levels of trust, consistent with H1.1. We similarly find a significant positive association for trust in experts.

We do not find a significant effect for social trust (H2) in this multivariate analysis, but stratification across groups could result in divergent vaccine behaviour. As we discussed previously, a strong theme in the focus groups was scepticism over death rates, inconsistent COVID-19 policies in the UK’s ”tier system” to ease restrictions in certain areas, and the unfair burden and punishment of those in the North, who have higher levels of socio-economic deprivation.

Those who perceived COVID-19 as a personal threat were almost two-and-a-half times more likely to express vaccine willingness than those who did not consider it a threat. As detailed below, a strong theme in the focus groups was that only the most vulnerable should get vaccinated, linked to ”herd immunity”, which the government used in early messaging and was widely discussed as pitting lockdowns versus no restrictions and achieving natural herd immunity [19]. This led some in the focus groups to believe that widespread infection would result in population immunity and, thus, little need for vaccination. The concept of herd immunity is complicated, and differs from the 70–80% vaccine herd immunity threshold, which is the proportion of the population required to block transmission, the level of which is related to vaccine efficacy and immunity duration [28]. Given the nuanced difference between herd immunity from COVID-19 infection and vaccine herd immunity, and the fact that the former was widely debated in the UK and internationally, it is unsurprising that there is confusion amongst the public. 

In response to H3, we find that holding conspiracy beliefs is a significant predictor of vaccine hesitancy. Furthermore, we find that individuals who obtain more information from the Internet are more willing to be vaccinated, but seeking online health information is widespread and heterogeneous. Only YouTube users were significantly less willing to be vaccinated, with a two-thirds likelihood of vaccine willingness compared to non-users. Instagram, TikTok, and Snapchat users were more hesitant, but when social media sources are disaggregated, our sample size is too small to draw firm conclusions. Facebook and Twitter users have slightly higher odds of vaccine willingness, but not significant at the 95% confidence level, and should therefore be judged with caution. Our findings linking YouTube users to COVID-19 vaccine hesitancy are novel, but in line with existing research on other vaccines. A study of YouTube vaccine content found that 65.5% of videos discouraged vaccine use, focussing on autism, undisclosed risks, adverse reactions, and alleged mercury content [29]. A 2017 analysis of 560 YouTube vaccine videos in Italy found that the majority of videos were negative, linking vaccines with autism and serious side effects [30]. Those who refused vaccines in the focus groups had low levels of trust in the government, and believed that the virus was man-made or a type of population control for certain groups. Individuals who were younger and had lower levels of education were also vaccine-hesitant.

Because vaccine distrust is proximate to our outcome variable (willingness to be vaccinated against COVID-19), as a sensitivity check, we also estimated the model excluding it as a predictor, as shown in Appendix A. This had minimal impact on the results, indicating that the effects of other attitudinal and behavioural predictors are robust to its inclusion. 

## 5. Discussion

Our findings offer further support to the evidence that trust and conspiracy beliefs predict vaccine hesitancy, both generally and for COVID-19 specifically [4,6,8,11,13]. They also highlight the importance of distinguishing between different *types* of social and institutional trust (i.e., trust in others, government, media, scientists/experts), in both theoretical and methodological terms. Like other studies, we find that trust in science and health organisations is important [13,16]. The perceived personal threat of COVID-19 and confidence in government handling of the pandemic are also associated with greater willingness to be vaccinated. In the bivariate analysis, we find some support for a relationship between social media use (of certain platforms: Snapchat, TikTok, YouTube, and Instagram) and increased vaccine hesitancy. Only the association for YouTube remains in the fully specified model, which could suggest that these findings reflect self-selection of particular subpopulations in social media usage. This highlights the potential for misinformation to impact on vaccine hesitancy through relatively unregulated and decentralized platforms [7,11,22].

Of demographic factors, age and education have the most robust associations with willingness to take the vaccine. The novel contribution of the paper to the fast-moving advances in this field comes both from its theorization of trust, mistrust, and distrust as distinct, extending recent studies [5], and its use of a mixed-methods approach. Insights from the focus groups serve to validate findings from the survey analysis, as well as shedding light on how individuals formulate judgments over the perceived safety of COVID-19 vaccines and their expressed willingness to be vaccinated. These were often founded not on ”irrational” thinking, but on understandable concerns about the (impressive) speed of vaccine development, or on misunderstandings of relevant concepts such as herd immunity. While some people are willing to entertain conspiratorial beliefs, these are rarely Manichean in nature, but rather attempts to make sense of fragmented and confusing information. 

We provide new evidence on how trust and information are linked with COVID-19 vaccine hesitancy, informing policy in key ways. Misinformation thrives where there is lack of trust in government, politics, and elites. A broader lesson is the need for authorities to communicate truthfully, transparently, and consistently. Over-promising, confusing messages, and blame rather than solving problems are faults of government and politicians that are best minimized—especially during times of crisis.

Personal perceived threat remains pivotal. With increased vaccinations and a drop in infections and deaths, individuals perceive lower threat. Our focus groups reveal that complacency emerges from a misunderstanding of “herd immunity”. What may seem to be irrational, conspiratorial judgements are often attempts to make sense of knowledge fragments accumulated during a fraught, complex, and rapidly evolving crisis. The public use a ”fast” and frugal model of intuitive thinking, using a mix of shortcuts and heuristics [31], which should be taken into account in communications. This fast and often emotional thinking during conditions of uncertainty can be clouded by social media, family, and friends, making it difficult for individuals to assess the relative importance of risks [32]. This inability to assess risk became clear in early 2021 in relation to the very rare blood clot disorders of 4 in 1 million, or 0.0004%, associated with the Oxford/AstraZeneca and Johnson & Johnson vaccines [32]. 

Since the Internet and social media are key sources for health information, governments should establish an engaging web presence in order to fill knowledge gaps [3]. Social media sites remain relatively unregulated, and since they do not operate as “publishers” that are forced to present balanced information, misinformation or conspiracy theories can quickly go “viral”. Some effective interventions could include advertisers boycotting their advertisements alongside harmful content [33]; companies can also check information, alter keyword searches, and redirect individuals to correct sources [3], ban overt conspiracy groups such as QAnon [3], balance viewpoints, flag misinformation, or rapidly remove content. Users can also be a source of misinformation correction, though the evidence remains inconclusive for COVID-19 so far [34]. Action also needs to be rapid; YouTube and Facebook removed “Plandemic”, but only after it was watched by millions [35]. Noting the source of information and forcing it to be traceable could be another measure. The most common sources of YouTube vaccine information are presented by non-expert individuals [29], suggesting that sites could flag or fact-check expertise of the video providers in order to help users gauge the accuracy or balance of information. However, expertise requires consensus, and in some rare cases, classic “experts”, such as medical doctors or even leading politicians, may not always provide accurate information. The viral YouTube film claiming that COVID-19 death certificates were manipulated was made by an anti-vax doctor, who is also a member of the Montana Health Board [36]. Lower COVID-19 vaccine uptake in some groups in the US has been linked to the former President Trump’s anti-vax views and tweets, which raised concerns about vaccine safety and are linked to belief in conspiracies [37]. 

This study is not without limitations, and invites extensions. We relied upon self-reports of media sources rather than objective logs. The data are cross-sectional, collected at a particular time point in the pandemic and global vaccination response, making it difficult to disentangle the causality of whether exposure to poor vaccine and health information shapes hesitancy, or a tendency to believe in conspiracies shapes information-seeking. Although our study is nationally representative, complemented by focus groups, the sample size remains small in a single country. Larger cross-national and longitudinal samples with multi-mode data gathering would be desirable. Nevertheless, with its mixed-methods design of a nationally representative survey with focus groups at the time of the initial vaccine rollout in the UK, exploring the core topics of trust and social media, this study provides a unique and vital window into contemporary COVID-19 vaccine hesitancy.

## Figures and Tables

**Figure 1 vaccines-09-00593-f001:**
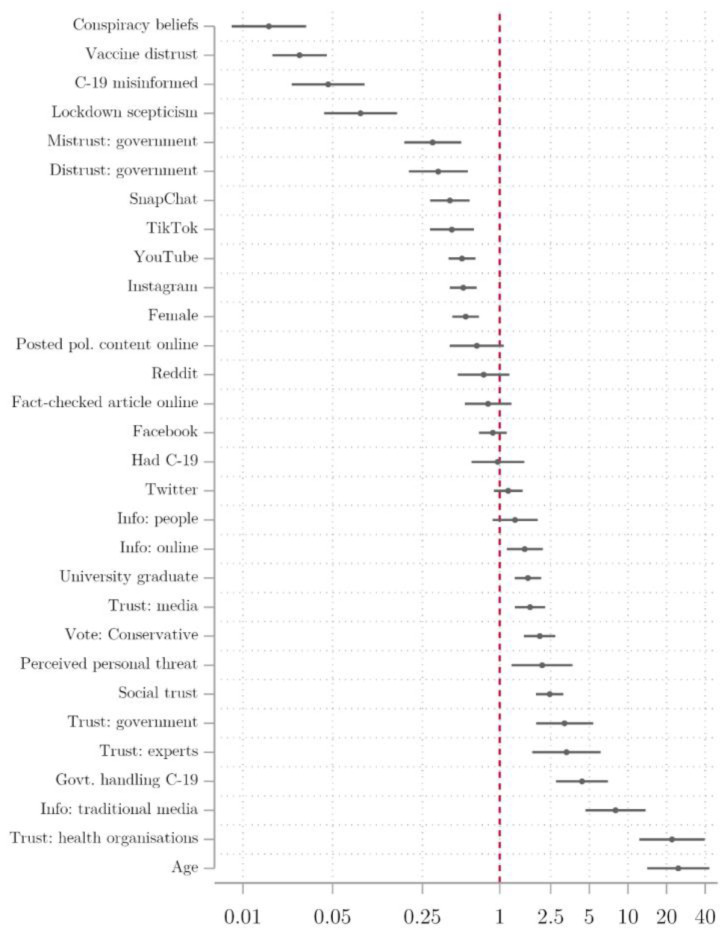
Bivariate logistic regression of vaccine willingness, odds ratios.

**Figure 2 vaccines-09-00593-f002:**
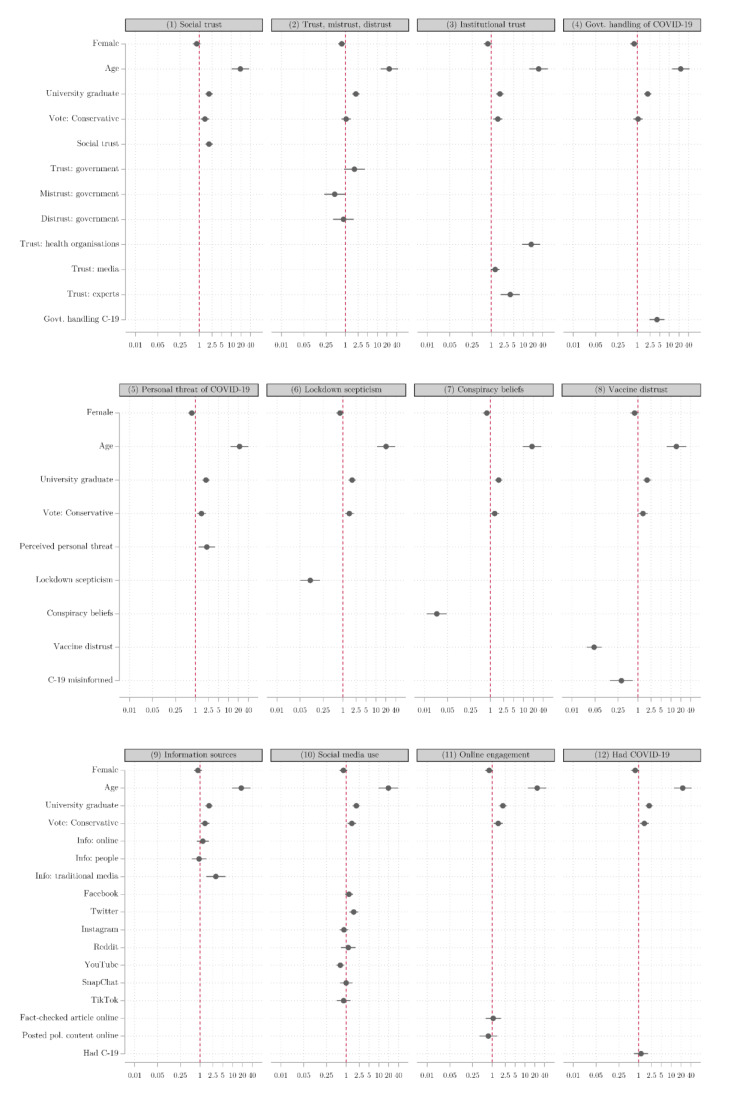
Multivariate logistic regression of vaccine willingness, odds ratios.

**Figure 3 vaccines-09-00593-f003:**
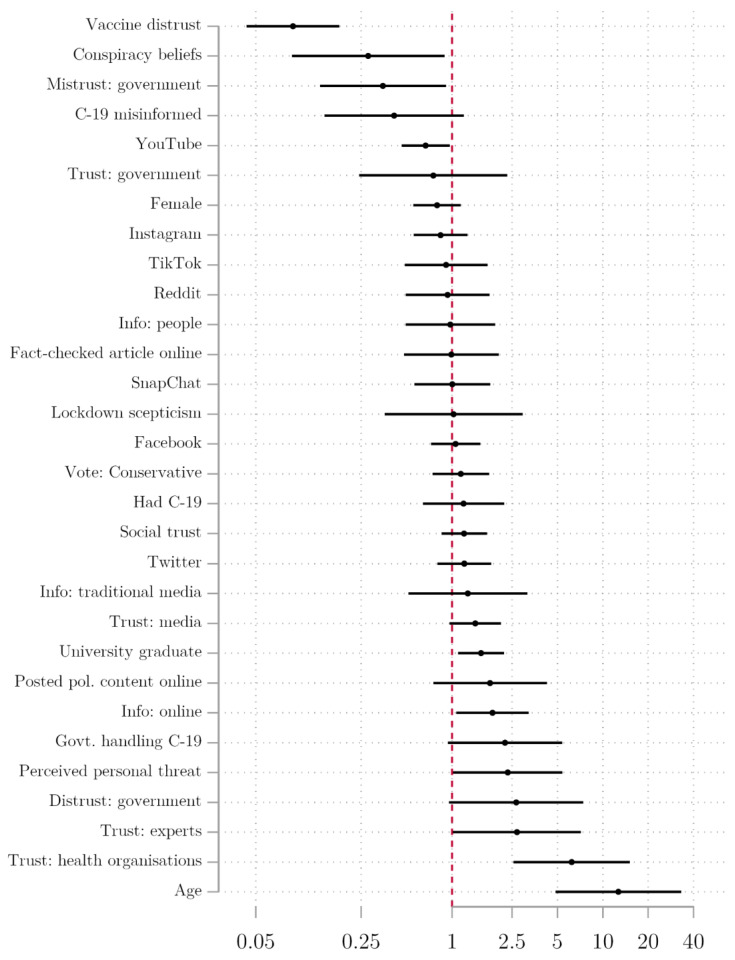
Odds ratios of determinants of vaccine willingness, combined logistic regression model.

**Table 1 vaccines-09-00593-t001:** Logistic regression estimates of vaccine willingness, odds ratios.

	(1)	(2)	(3)	(4)
Trust				
Social trust	1.273	1.254	1.228	1.205
	(0.923–1.754)	(0.901–1.745)	(0.880–1.714)	(0.852–1.705)
Trust: government	0.696	0.705	0.696	0.751
	(0.242–2.005)	(0.240–2.075)	(0.234–2.070)	(0.243–2.324)
Mistrust: government	0.400	0.381	0.395	0.349
	(0.164–0.977) *	(0.153–0.949) *	(0.157–0.996) *	(0.133–0.913) *
Distrust: government	2.420	2.363	2.180	2.665
	(0.956–6.125)	(0.896–6.227)	(0.817–5.818)	(0.958–7.415)
Trust: health organisations	6.154	6.294	6.019	6.218
	(2.735–13.846) ***	(2.715–14.592) ***	(2.569–14.105) ***	(2.560–15.104) ***
Trust: media	1.394	1.306	1.349	1.428
	(0.972–2.001)	(0.901–1.894)	(0.926–1.965)	(0.965–2.112)
Trust: experts	1.958	1.716	1.718	2.695
	(0.810–4.736)	(0.697–4.225)	(0.692–4.264)	(1.013–7.171) *
COVID-19/Vaccines				
Government handling of COVID-19	2.323	2.131	2.204	2.249
	(1.020–5.292) *	(0.920–4.939)	(0.945–5.142)	(0.939–5.389)
Perceived personal threat of COVID-19	2.329	2.221	2.216	2.344
	(1.073–5.053) *	(1.004–4.915) *	(0.993–4.942)	(1.016–5.405) *
Lockdown scepticism	1.181	0.906	0.888	1.026
	(0.446–3.128)	(0.331–2.481)	(0.322–2.447)	(0.358–2.943)
Conspiracy beliefs	0.307	0.292	0.294	0.279
	(0.106–0.891) *	(0.097–0.881) *	(0.096–0.897) *	(0.087–0.894) *
Vaccine distrust	0.083	0.090	0.091	0.088
	(0.043–0.159) ***	(0.046–0.177) ***	(0.046–0.180) ***	(0.044–0.179) ***
COVID-19 misinformed	0.445	0.476	0.490	0.414
	(0.170–1.167)	(0.178–1.275)	(0.181–1.324)	(0.143–1.199)
Demographics				
Had COVID-19	1.257	1.317	1.280	1.193
	(0.688–2.298)	(0.712–2.434)	(0.692–2.369)	(0.641–2.222)
Female	0.731	0.797	0.787	0.797
	(0.528–1.012)	(0.570–1.116)	(0.554–1.116)	(0.555–1.145)
Age	14.897	13.781	10.341	12.684
	(6.862–32.341) ***	(6.126–31.002) ***	(4.152–25.756) ***	(4.848–33.189) ***
Graduate	1.701	1.572	1.574	1.560
	(1.239–2.334) **	(1.133–2.181) **	(1.127–2.198) **	(1.098–2.215) *
Supports Conservative Party	1.210	1.202	1.167	1.146
	(0.806–1.818)	(0.794–1.821)	(0.768–1.773)	(0.744–1.765)
Media/Information				
Information sources: online		1.716	1.664	1.860
		(1.028–2.864) *	(0.985–2.809)	(1.071–3.232) *
Information sources: people		1.022	1.028	0.976
		(0.535–1.951)	(0.536–1.973)	(0.492–1.938)
Information sources: traditional		1.565	1.557	1.276
		(0.666–3.676)	(0.654–3.702)	(0.514–3.167)
Social media use: Facebook			1.131	1.058
			(0.787–1.625)	(0.726–1.541)
Social media use: Twitter			1.331	1.206
			(0.902–1.964)	(0.799–1.819)
Social media use: Instagram			0.832	0.840
			(0.561–1.236)	(0.557–1.268)
Social media use: Reddit			1.004	0.935
			(0.540–1.867)	(0.493–1.774)
Social media use: Youtube			0.672	0.669
			(0.472–0.956) *	(0.463–0.968) *
Social media use: Snapchat			0.998	1.006
			(0.567–1.755)	(0.563–1.796)
Social media use: TikTok			0.951	0.915
			(0.512–1.764)	(0.487–1.719)
Fact-checked an article online				0.992
				(0.481–2.043)
Posted political content online				1.790
				(0.752–4.260)
N	1348	1316	1316	1261
Pseudo R-squared	0.28	0.29	0.29	0.30

Notes: * *p* < 0.05; ** *p* < 0.01; *** *p* < 0.001 (95% confidence intervals in parentheses).

## Data Availability

We commissioned Ipsos MORI to conduct a nationally representative online survey of 1476 adults in the UK from 12 December to 18 December 2020, and 5 focus groups conducted between 30 November and 7 December 2020. The data presented in this study are available on request from the corresponding author.

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
