# Peer review of "Lack of Trust, Conspiracy Beliefs, and Social Media Use Predict COVID-19 Vaccine Hesitancy"

_vaccines, 2021, doi:10.3390/vaccines9060593_

Round 1

Reviewer 1 Report

Thank you for the opportunity to review this very interesting paper, with which I was already familiar. It is timely, relevant to the scope of the journal, and interesting.

I recommend it should be accepted, but raise the following points.

  1. Flagging of the effect of 'echo chambers' in the title and throughout the paper seems like an over-reach. Of course we know that people inhabit different informational ecosystems online, sometimes to an extreme degree. We know that the online world has important offline impacts. And yet the only significant finding in the models is rather slight and relates to YouTube. The description of how viewing begets viewing on YouTube is engaging, but quite a long way divorced from what the analysis can actually tell us. I also thought that the description/analysis of the focus group conversations in the SI a missed opportunity with regard to detail on social media consumption. A clearer acknowledgement of likely selection effects is required. I strongly advise that the title is amended to remove the reference to 'echo chambers', which positions the paper wrongly.
  2. The variable names in Figure 1 are near-impossible to read.
  3. I could not find the supplementary information via the journal submission system and referred instead to that on the MedRXiv preprint repository.
  4. I apologize for making novice points but the models are explaining attitudes by attitudes to a level beyond my own comfort level. There could be a cleared account of how mistrust, trust and distrust form different (ideally orthogonal) dimensions of trust. There's a question of balance between informative analysis of associations, particularly of value to the policy world, and focus on parsimony/causal identification. The high pseudo R-squared figures in combination with wide confidence intervals suggest some specification issues.
  5. Some missing references, notably to Loomba et al 2021 https://www.nature.com/articles/s41562-021-01056-1 and rather fewer references to the literature on vaccine hesitancy, and vaccine hesitancy online, than I might have expected. It's worth citing Bode and Vraga on social media interventions in this space. Dubé on vaccine hesitancy in general requires a cite. Kahneman is always engaging to see, but surely not central to the established body of material in this space already, including the growing body of work on coronavirus vaccine hesitancy which is also underplayed. More careful identification of what is new in this paper, compared with emerging work, would be useful for the field. 
  6. I'm not sure there is a clear reference in the main paper to the number of participants in the focus groups (which seems to be 29). Some of the focus group findings are interesting but the small (understandably so) N and lack of space to get into detail meant they were a little thin.

Author Response

Dear Editors,

Thank you for the opportunity to resubmit our manuscript to Vaccines. The reviewers offered a range of valuable insights and important suggestions for revision, which we have addressed carefully and systematically in the revised manuscript. At the reviewers’ suggestion, we have substantially expanded our discussion of the empirical material (including analysis of both the survey data and focus groups), in a way that we believe makes the paper a more novel and extensive contribution to the field. In this memorandum we outline the specific revisions made in response to each reviewer below, and here first summarise the main points that extend across the different reviews.

  1. We have expanded the article to more comprehensively discuss the survey analysis and focus group material. The paper was originally prepared as a research note, with more extensive supplementary materials - which we thought we had included with the original submission which included more extensive details of the focus groups. We are very happy to offer an expanded version. Doing so enables us to provide clarification on many of the issues raised by the reviewers.
  2. We have expanded the literature review, including references suggested (Loomba et al. 2021; Dubé et al. 2013) as well as other recent relevant publications (e.g. Sturgis et al. 2021).
  3. We have specified our main hypotheses and ensured the discussion more directly relates to these.
  4. We have more comprehensively detailed our (quantitative and qualitative) methodology, including a more extensive description of the survey sample and focus group design.
  5. We have improved presentation of the figures (making labels more legible) and statistical analyses, ensuring the discussion consistently refers to odds ratios (rather than predicted probabilities) .
  6. We have been more careful in discussion of how the study relates to social media, removing references to 'echo chambers' in the title and related discussion in the main text.
  7. We have added a summary of our findings to the closing discussion so that the implications drawn are clearly related to the empirical findings. We would be happy to make further revisions to this section, but feel that reflecting on wider policy debates is an important contribution of the paper as a whole.

Reviewer #1

We thank the reviewer for their positive response and helpful comments and suggestions, which have provided us a clear focus for revising the manuscript.

Following the advice of the reviewer we have removed ‘echo chambers’ from the title and references to the idea, recognising that our findings do not support claims in this regard.

By extending the paper we have provided greater detail of the focus group design and analysis, which we hope provides additional insights into importance of the information environment of individuals. We also note selection effects of social media consumption.

We have revised Figure 1 so that the text size of labels should be clearer to read.

We have added a note on the difference between the trust, mistrust and distrust dimensions of our survey measures, including an additional reference to support for use of these distinct constructs (page 5).

We have added references to Loomba et al. (2021) and Dube et al (2013), plus other recent relevant publications (e.g. Sturgis et al. 2021), on vaccine hesitancy, plus Bode and Vraga (2021) on social media interventions, and thank the reviewer for prompting us to do this.

We have sought in the conclusion to specify the contribution that our analysis offers in relation to vaccine hesitancy generally and coronavirus vaccine hesitancy specifically. This notes the insights afforded by the mixed method approach, for understanding the predictors of vaccine hesitancy as well as the way that individuals make judgments regarding vaccine safety.

Reviewer 2 Report

The present manuscript is well-structured, well-written and easy to understand on the topics of Lack of trust and social media echo chambers predict COVID- 19 vaccine hesitancy.

I consider all sections are solid and very interesting for the readers.

As COVID-19 vaccines are rolled out across the world, there are growing concerns about the role that trust, belief in conspiracy theories and spread of misinformation through social media impact vaccine hesitancy.

I have no any other comments on this manuscript, I strong suggest the editor accept at the current form ASAP and could let it rapid online to help us to fighting the COVID-19.

Author Response

Dear Editors,

Thank you for the opportunity to resubmit our manuscript to Vaccines. The reviewers offered a range of valuable insights and important suggestions for revision, which we have addressed carefully and systematically in the revised manuscript. At the reviewers’ suggestion, we have substantially expanded our discussion of the empirical material (including analysis of both the survey data and focus groups), in a way that we believe makes the paper a more novel and extensive contribution to the field. In this memorandum we outline the specific revisions made in response to each reviewer below, and here first summarise the main points that extend across the different reviews.

  1. We have expanded the article to more comprehensively discuss the survey analysis and focus group material. The paper was originally prepared as a research note, with more extensive supplementary materials - which we thought we had included with the original submission which included more extensive details of the focus groups. We are very happy to offer an expanded version. Doing so enables us to provide clarification on many of the issues raised by the reviewers.
  2. We have expanded the literature review, including references suggested (Loomba et al. 2021; Dubé et al. 2013) as well as other recent relevant publications (e.g. Sturgis et al. 2021).
  3. We have specified our main hypotheses and ensured the discussion more directly relates to these.
  4. We have more comprehensively detailed our (quantitative and qualitative) methodology, including a more extensive description of the survey sample and focus group design.
  5. We have improved presentation of the figures (making labels more legible) and statistical analyses, ensuring the discussion consistently refers to odds ratios (rather than predicted probabilities) .
  6. We have been more careful in discussion of how the study relates to social media, removing references to 'echo chambers' in the title and related discussion in the main text.
  7. We have added a summary of our findings to the closing discussion so that the implications drawn are clearly related to the empirical findings. We would be happy to make further revisions to this section, but feel that reflecting on wider policy debates is an important contribution of the paper as a whole.

Reviewer #2

We thank the reviewer for their positive assessment of the original manuscript and hope that they endorse the extensions that provide additional detail in relation to the existing theoretical and methodological framework.

Reviewer 3 Report

I read with interest the paper by Jennings and colleagues starting from the title and abstract which clearly address a very important topic like the one regarding trust in vaccines and confidence.

Nevertheleess I should clearly specify that the paper has some major flaws including the fact that it lacks completely a clear methodological section.

Since, in my opinion, it is very difficult to list all the flaws I will make a couple of examples of the major ones:

- How are the groups created? How are them defined on the basis of the hypotesis you want to test?

- What are the main characteristics of the sample? When I teach how to write a paper to my students or residents I always point out that a "Table 1" with the characteristics of the sample selected should always be pointed out. From this paper it is impossible to understand even if this 1.476 people of the "representative sample" of this survey were men or women, child or adults. Obviously this clearly prevents me from judging clearly the results of your table 1 or figure 1.

- also in the results section is quite impossible to read/interpret the results presented in figure 1. Which, on the other way round, is redundant with table 1.

I am really sorry to say that, in my opinion, in this form this paper is not acceptable in Vaccines. This paper must be rewritten completely following a clear methodology and template of a scientific study.

Author Response

Dear Editors,

Thank you for the opportunity to resubmit our manuscript to Vaccines. The reviewers offered a range of valuable insights and important suggestions for revision, which we have addressed carefully and systematically in the revised manuscript. At the reviewers’ suggestion, we have substantially expanded our discussion of the empirical material (including analysis of both the survey data and focus groups), in a way that we believe makes the paper a more novel and extensive contribution to the field. In this memorandum we outline the specific revisions made in response to each reviewer below, and here first summarise the main points that extend across the different reviews.

  1. We have expanded the article to more comprehensively discuss the survey analysis and focus group material. The paper was originally prepared as a research note, with more extensive supplementary materials - which we thought we had included with the original submission which included more extensive details of the focus groups. We are very happy to offer an expanded version. Doing so enables us to provide clarification on many of the issues raised by the reviewers.
  2. We have expanded the literature review, including references suggested (Loomba et al. 2021; Dubé et al. 2013) as well as other recent relevant publications (e.g. Sturgis et al. 2021).
  3. We have specified our main hypotheses and ensured the discussion more directly relates to these.
  4. We have more comprehensively detailed our (quantitative and qualitative) methodology, including a more extensive description of the survey sample and focus group design.
  5. We have improved presentation of the figures (making labels more legible) and statistical analyses, ensuring the discussion consistently refers to odds ratios (rather than predicted probabilities) .
  6. We have been more careful in discussion of how the study relates to social media, removing references to 'echo chambers' in the title and related discussion in the main text.
  7. We have added a summary of our findings to the closing discussion so that the implications drawn are clearly related to the empirical findings. We would be happy to make further revisions to this section, but feel that reflecting on wider policy debates is an important contribution of the paper as a whole.

Reviewer #3

The reviewer was dissatisfied with the level of methodological detail provided in the original submission (this was due to the research note format used, with additional information included in the supplementary material). We hope that the considerably extended version of the paper, now providing substantially more detail regarding both the survey design and sample, and the focus group sample and analysis, addresses the concerns of the reviewer. As regards the survey, the sample is designed to be nationally representative of British adults, and further information has been provided in the manuscript on page 3. We also explain the rationale for choice of the focus group samples on page 3.

We have revised Figure 1 so that the text size of labels should be clearer to read. While we agree that this duplicates the findings from Table 1, we feel that this visualisation allows for a more accessible reading of the empirical results (now that the labels are more legible).

Reviewer 4 Report

This is potentially a very important study: the combination of focus groups and survey research is innovative, and to be commended. The survey research appears robust and well reported, although I found it confusing that some effect sizes were reported as percentage probabilities in the text itself, while effect sizes were only reported as odds ratios in the table and the visualisation: if some percentage probabilities are to be reported in the text, I would prefer them to be reported for all effects in the tables; moreover, it wasn’t clear whether the percentage probabilities quoted in the text reflected bivariate relationships or were adjusted for controls, and – if the latter – it also wasn’t clear which of the four models they related to. Perhaps it would be easier to use odds ratios throughout?

Except in this minor regard, I am very happy for the part of the article that reports on the survey to be published as it is. However, the reporting of the focus group research was rather perfunctory: there was a reference to ‘detailed focus group results in the SI’, which unfortunately I wasn’t able to access; apart from that, there were only three sentences devoted to findings of that aspect of the research (one of which was picked up again in the Discussion). I initially assumed that such extreme brevity was a consequence of the journal’s word limit, but the journal website suggests that papers of up to 10000 words may be submitted without obtaining special permission. I would recommend either removing all reference to the focus group research and publishing it in a separate article, or otherwise substantially expanding discussion of the methods and findings of the focus group research within the current article. Obviously, the former course of action would be an easier route to acceptance for the current article, although it would mean that it was no longer a mixed methods study.

The literature review could also do with expanding, as it is rather brief and does not cover the range of research that has been done on these topics since the beginning of the current pandemic. In particular, the statement that ‘[c]onspiracy and anti-vaxx beliefs and low trust in institutions is associated with a greater reliance on social media for health information, but research until now has primarily used small selective samples (e.g. MTurk)’ is backed up with citations of just two studies, both from before the pandemic. There have been a number of survey-based studies published since the middle of 2020 looking at the relationship between informational reliance on social media and one or more of: COVID-19 conspiracy beliefs, COVID-19 health protective behaviours, trust, and vaccine intentions. These include studies carried out both in the UK and elsewhere, and many of them have used large samples and even representative samples. Some may have appeared since the manuscript was submitted, but coverage would nonetheless be expected in the literature review. (I note that two are currently cited in the Materials and Methods section, so with regard to those, it would just be a matter of citing in the literature review also.) There was also no coverage of earlier focus group studies of vaccine hesitancy, whether with regard to COVID-19 or with regard to other infectious diseases, although I am aware of a number that have been published. Similarly, while there was a reference to a ‘2017 analysis of 560 YouTube vaccine videos in Italy’ (in the Results section rather than the literature review), there was no mention of similar studies that have been carried out, some since the beginning of the COVID-19 pandemic, looking at YouTube content in English (which is presumably more relevant to the current study’s findings). In this context, I should also note that at least two published studies have found a particularly strong correlation between YouTube usage and COVID-19 vaccine hesitancy, which is reported as a ‘novel’ finding of the current study (a third relatedly found a particularly strong relationship between YouTube usage and COVID-19 conspiracy beliefs).

The discussion section of the article seemed based more on a reading of the existing literature than on the findings reported in the body of the article itself. I would suggest that most of it could be moved directly into the introduction, and that a new discussion section should be written which directly relates the findings of the research reported in the article to the findings of other studies that it could be considered to replicate.

Author Response

Dear Editors,

Thank you for the opportunity to resubmit our manuscript to Vaccines. The reviewers offered a range of valuable insights and important suggestions for revision, which we have addressed carefully and systematically in the revised manuscript. At the reviewers’ suggestion, we have substantially expanded our discussion of the empirical material (including analysis of both the survey data and focus groups), in a way that we believe makes the paper a more novel and extensive contribution to the field. In this memorandum we outline the specific revisions made in response to each reviewer below, and here first summarise the main points that extend across the different reviews.

  1. We have expanded the article to more comprehensively discuss the survey analysis and focus group material. The paper was originally prepared as a research note, with more extensive supplementary materials - which we thought we had included with the original submission which included more extensive details of the focus groups. We are very happy to offer an expanded version. Doing so enables us to provide clarification on many of the issues raised by the reviewers.
  2. We have expanded the literature review, including references suggested (Loomba et al. 2021; Dubé et al. 2013) as well as other recent relevant publications (e.g. Sturgis et al. 2021).
  3. We have specified our main hypotheses and ensured the discussion more directly relates to these.
  4. We have more comprehensively detailed our (quantitative and qualitative) methodology, including a more extensive description of the survey sample and focus group design.
  5. We have improved presentation of the figures (making labels more legible) and statistical analyses, ensuring the discussion consistently refers to odds ratios (rather than predicted probabilities) .
  6. We have been more careful in discussion of how the study relates to social media, removing references to 'echo chambers' in the title and related discussion in the main text.
  7. We have added a summary of our findings to the closing discussion so that the implications drawn are clearly related to the empirical findings. We would be happy to make further revisions to this section, but feel that reflecting on wider policy debates is an important contribution of the paper as a whole.

Reviewer #4

We thank the reviewer for their positive assessment of the paper and their helpful comments.

Following the reviewer’s recommendation, we have rewritten the analysis so it is consistent in referring to the odds ratios – as reported in the figures and tables.

We are delighted to provide a more extensive description of the focus group design and analysis in the paper (and apologise that the supplementary information was not available for the original submission). We hope that the reviewer finds this extension of the analysis provides important additional information regarding the factors influencing vaccine hesitancy.

Following the reviewer’s suggestion, we have also expanded the literature review (pages 1 to 2), including relevant studies on the relationship between social media, trust, conspiracy beliefs and vaccine intentions (including those specifically relating to social media and disinformation, YouTube and vaccine hesitancy). As the reviewer notes, this is a fast-moving field but we are pleased to be able to update our review of the latest advances in this field.

Lastly, we have added a summary of our empirical findings to the discussion section, before discussing how those findings relate to other studies in this area. We have left some of the discussion of wider implications of the study, but hope that the reviewer agrees that there is now greater balance to this section.

Round 2

Reviewer 3 Report

I reread with renewed interest the paper by Jennings and colleagues, finding it really improved, even following what had been perhaps my rather "tranchant" judgments.I would like to congratulate the authors for the work done and for the substantial improvement that now makes this work really worthy of publication.